# Perinatal factors affecting platelet parameters in late preterm and term neonates

**Hayato Go**[1]*, **Hitoshi Ohto**[2], **Kenneth E. Nollet**[3], **Nozomi Kashiwabara**[1], **Mina Chishiki**[1], **Masato Hoshino**[1], **Kei Ogasawara**[1], **Yukihiko Kawasaki**[4], **Nobuo Momoi**[1], **Mitsuaki Hosoya**[1]

1 Department of Pediatrics, Fukushima Medical University School of Medicine, Fukushima, Japan, 2 Department of Advanced Cancer Immunotherapy, Fukushima Medical University School of Medicine, Fukushima, Japan, 3 Department of Blood Transfusion and Transplantation Immunology, Fukushima Medical University School of Medicine, Fukushima, Japan, 4 Department of Pediatrics, Sapporo Medical University School of Medicine, Hokkaido, Japan

* gohayato2525@gmail.com

## Abstract

Platelets parameters including platelet count (PLT), plateletcrit (PCT), mean platelet volume (MPV) and platelet distribution width (PDW) are associated with various physiological and pathological functions in various disease. However, few studies have addressed whether perinatal factors may be associated with platelet parameters at birth in a large cohort of late preterm and term neonates. The aim of this study to investigate perinatal factors affecting platelet parameters in late preterm and term neonates. We retrospectively investigated platelet parameters including PLT, PCT, MPV, and PDW on the first day of life in 142 late preterm and 258 term neonates admitted to our NICU from 2006 through 2020. PLT, MPV, PCT, PDW on Day 0 did not significantly differ between the two groups. In term neonates, multivariate analysis revealed that PCT correlated with being small for gestational age (SGA) ($\beta$ = -0.168, $P$ = 0.006), pregnancy induced hypertension (PIH) ($\beta$ = -0.135, $P$ = 0.026) and male sex ($\beta$ = -0.185, $P$ = 0.002). PLT was associated with SGA ($\beta$ = -0.186, $P$ = 0.002), PIH ($\beta$ = -0.137, $P$ = 0.024) and male sex ($\beta$ = -0.166, $P$ = 0.006). In late preterm neonates, multivariate analysis revealed that PLT were associated with PIH, whereas no factors associated with PDW and MPV were found. In all patients studied, chorioamnionitis (CAM) was significantly associated with MPV (CAM = 10.3 fL vs. no CAM = 9.7 fL, $P$<0.001). Multivariate analysis showed that SGA, male sex and PIH were associated with PCT and PLT. This study demonstrates that different maternal and neonatal complications affect platelet parameters in late preterm and term neonates.

## Introduction

Platelets parameters including platelet count (PLT), plateletcrit (PCT), mean platelet volume (MPV) and platelet distribution width (PDW) are associated with various physiological and pathological functions in various disease [1–3]. Platelet production is a complex process arising from the proliferation and differentiation of megakaryocytes under the stimulating

**Data Availability Statement:** All relevant data are within the manuscript and its Supporting information files.

**Funding:** This research proceeded without the benefit of grant money or any other external

financial support. Accordingly, funders had no role in study design, data collection and analysis, decision to publish, or preparation of the manuscript.

**Competing interests:** The authors have declared that no competing interests exist.

influence of thrombopoietin. Platelet function in premature neonates is immature. Previous research indicated that low birth weight infants were at almost 2.5-fold increased risk for thrombocytopenia [4]. On the other hand, MPV, PDW, PCT are associated with neonatal disease such as sepsis, fungal infection and intraventricular hemorrhage [5–8]. Among extremely low birth weight neonates born to mothers with preeclampsia, the MPV/platelet count (PLT) ratio at birth significantly correlated with mortality [9]. Also, changes of PDW during the neonatal period were associated with thrombocytopenia and with neonatal sepsis in very low birth weight infants [10]. We previously reported that higher MPV correlates with mortality among those born at < 32weeks' gestation [11].

On the other hand, Alicja et al reported decreased PCT and PLT in late preterm neonates compared with term neonates [12]. However, few studies have addressed whether perinatal factors may be associated with platelet parameters (PCT, PLT, MPV and PDW) at birth in a large cohort of late preterm and term neonates. Recently, some studies have been associated maternal complication such as premature rupture of membranes (PROM), chorioamnionitis (CAM), and pregnancy induced hypertension (PIH) were associated with platelet parameters [13–15]. Therefore, we hypothesized that platelet parameters such as MPV, PCT and PDW could be affected by perinatal factors. The objective of this study was to investigate the factors affecting platelet parameters at birth in late preterm and term neonates.

## Materials and methods

### Study design and population

This retrospective, single center, cohort study examined records from 2006 through 2020 at the neonatal intensive care unit (NICU) of Fukushima Medical University Hospital. The study protocol was approved by our institutional review board, The Ethics Committee of Fukushima Medical University, Fukushima, Japan. After deliberation, the board decided that no written consent was needed. Of 1789 neonates admitted to our NICU between January 2006 and Mar 2020, there were 400 born at term or late preterm (weeks 35–36). Late preterm neonates and full term neonates numbered 142 and 258, respectively (Fig 1). Exclusion criteria were congenital anomalies, neonatal alloimmune thrombocytopenia, and neonates not tested within 12 hours of birth.

### Platelet parameter measurements

Blood samples were collected through peripheral venipuncture to measure PLT, PCT, MPV and PDW of each newborn. Complete blood counts were measured using a Sysmex CS-5100 coagulation analyzer (Sysmex, Kobe, Japan) on admission.

### Prenatal and postnatal risk factors

Platelet parameters were compared with demographic variables including gender, birth weight (BW), and gestational age (GA). Furthermore, we considered pregnancy-induced hypertension (PIH), chorioamnionitis (CAM, defined clinically, with histopathological confirmation in pre-term placentas) [16]. premature rupture of membranes (PROM), and placental abruption (PA) as possible prenatal risk factors, and small for gestational age (SGA), respiratory distress syndrome (RDS), and Apgar scores as possibly associated with platelet parameters. Infants whose birth weight and height were below the 10th percentile of the normal curve at each GA were classified as SGA.

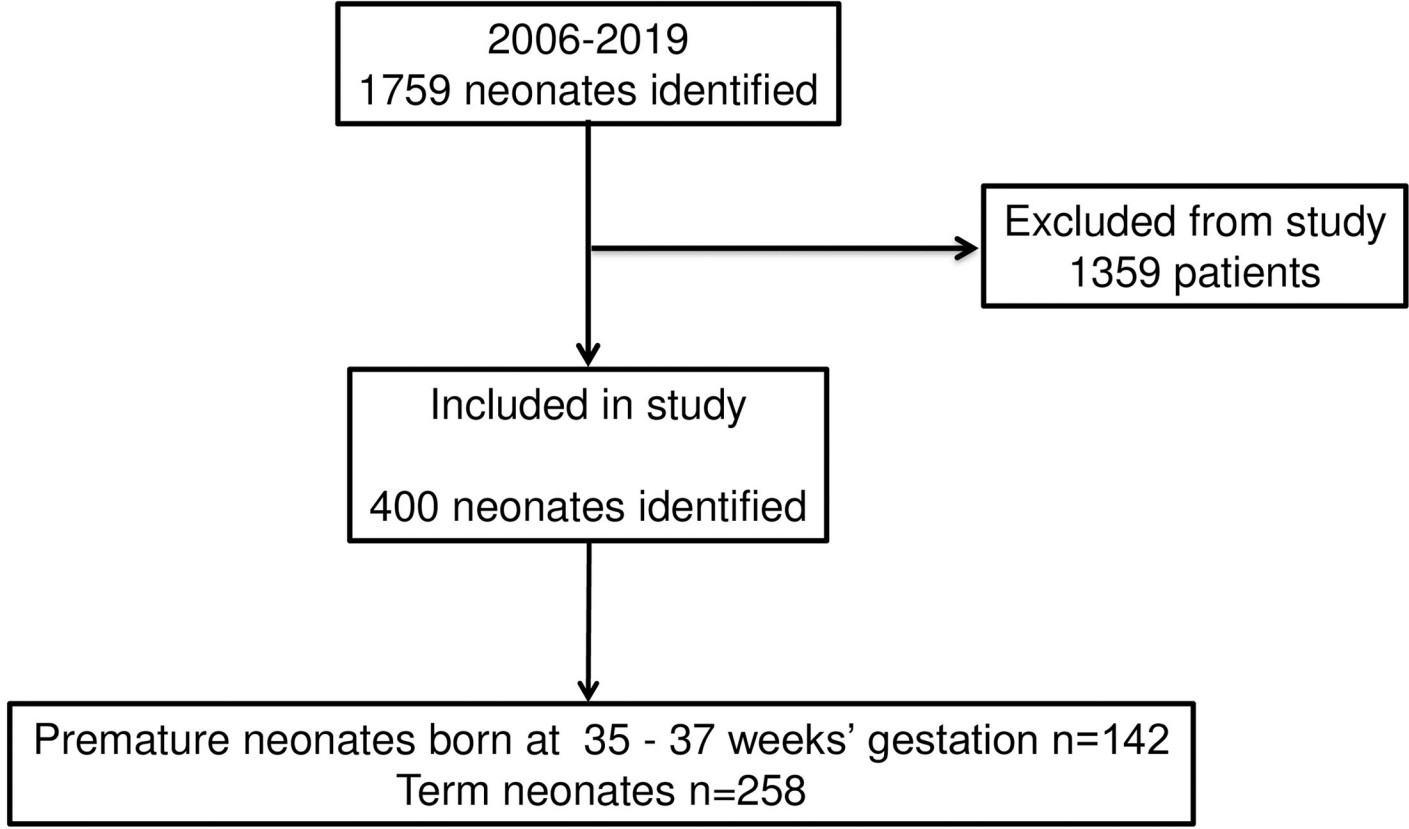

**Fig 1. Flowchart showing enrollment of study subjects.**

### Statistical analysis

Platelet parameters from medical records were rendered into median and interquartile range (IQR) as non-normally distributed continuous variables. Differences in categorical variables and continuous variables were assessed for significance using Chi-square and Mann–Whitney U test. Likewise, correlations between BW, GA, and Apgar scores versus platelet parameters were also investigated using Spearman's rank correlation ($r$) with $P < 0.05$ considered to be statistically significant. Moreover, the variables at $P < 0.05$ in univariate analysis were entered into a multiple regression analysis to identify independent prognostic factors.

Data analysis was performed with SPSS for Mac, release 25.0 (SPSS, Chicago, IL) and Prism 8 (GraphPad Software, San Diego, CA).

### Results

Table 1 shows platelet parameters in late preterm and term neonates. There are no significant differences in platelet parameters between late preterm and term neonates. The median of sampling time was 82 min as shown in Fig 2. Fig 3A and 3B show no significant correlation between BW, GA and PLT. Likewise, Fig 3C and 3D show no significant correlation between BW, GA and PCT in late preterm and term neonates.

Tables 2 and 3 outline the results of univariate and multivariate analyses for the correlation between perinatal factors and platelet parameters in preterm and term neonates, respectively.

**Table 1. Platelet parameters in late preterm and term neonates.**

|  | Late preterm (n = 142) Median [IQR] | Term (n = 258) Median [IQR] | p-value |
|---|---|---|---|
| GA (weeks) | 36.0 [35.4–36.4] | 38.2 [37.5–39.3] | <0.001 |
| BW (grams) | 2180 [1802–2602] | 2758 [2463–3106] | <0.001 |
| SGA n (%) | 57 (39.8%) | 52 (20.2%) | <0.001 |
| PCT (%) | 0.25 [0.21–0.31] | 0.26 [0.21–0.30] | 0.661 |
| PDW | 10.9 [10.0–12.7] | 10.9 [10.2–12.2] | 0.810 |
| MPV (fL) | 9.9 [9.2–10.3] | 9.7 [9.3–10.3] | 0.792 |
| PLT (×10³/μL) | 26.6 [22.2–32.3] | 26.4 [21.5–31.4] | 0.807 |

GA, gestational age; BW, birth weight; PCT, plateletcrit; PDW, platelet distribution width; MPV, mean platelet volume; PLT, platelet count; IQR, median interquartile range.

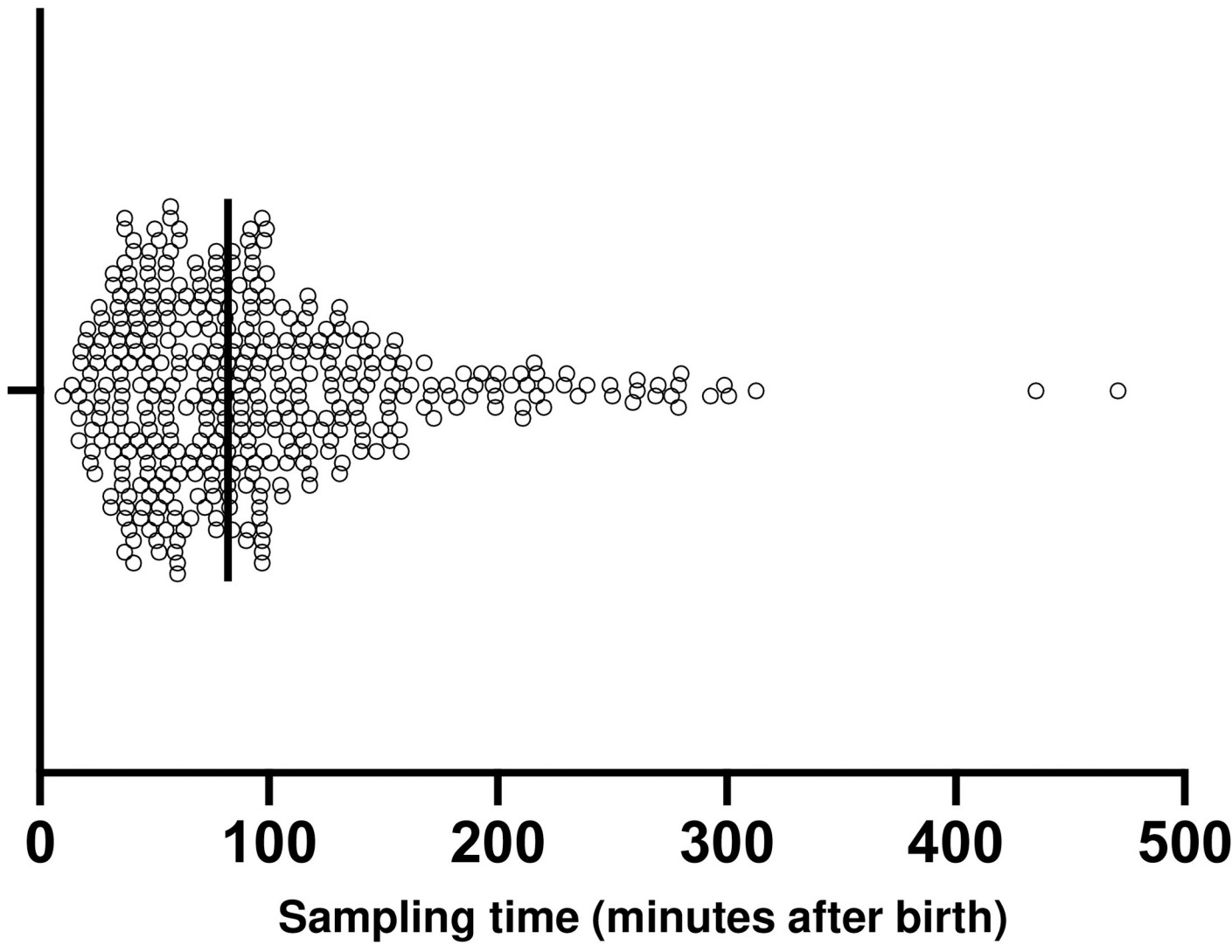

**Fig 2. Peripheral blood sampling time distribution.**

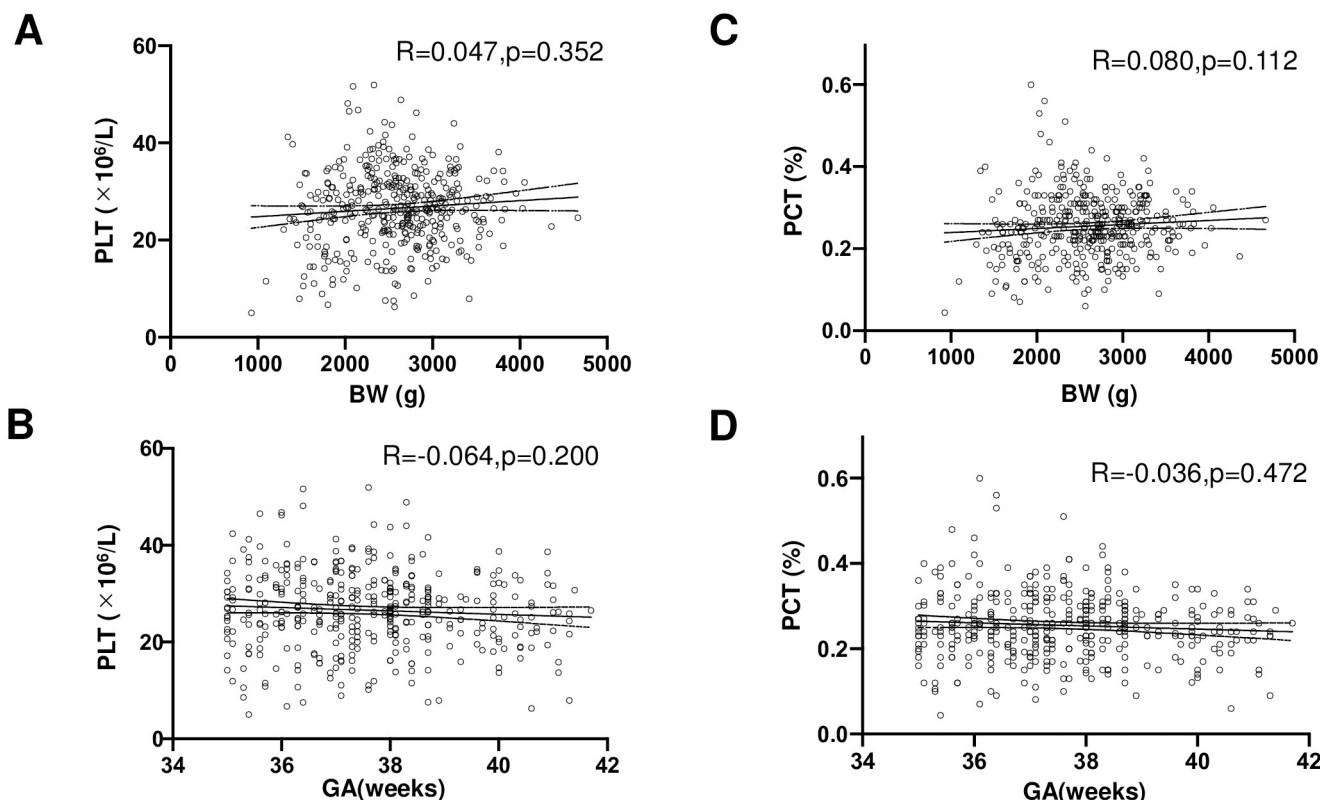

**Fig 3. Correlation between platelet count, plateletcrit, GA, and BW.** R is Spearman's correlation coefficient.

As shown in Table 2, in term neonates, MPV was associated with CAM (CAM = 10.4 fL vs. no CAM = 9.7 fL, $P$ = 0.005). Furthermore, PCT was associated with male sex (male = 0.24% vs. female = 0.27%, $P$ = 0.005), SGA (SGA = 0.23% vs. no SGA = 0.26%, $P$ = 0.024) and PIH (PIH = 0.19% vs. no PIH = 0.50%, $P$ = 0.009), respectively. PLT was also associated with male sex (male = $25.9 \times 10^3/\mu$L vs. female = $28.8 \times 10^3/\mu$L, $P$ = 0.009), SGA (SGA = $24.8 \times 10^3/\mu$L vs. no SGA = $26.7 \times 10^3/\mu$L, $P$ = 0.023) and PIH (PIH = $17.4 \times 10^3/\mu$L vs. no PIH = $26.5 \times 10^3/\mu$L, $P$ = 0.007), respectively. Multivariate analysis revealed that PCT correlated with male sex ($\beta$ = -0.185, $P$ = 0.002), SGA ($\beta$ = -0.168, $P$ = 0.006) and PIH ($\beta$ = -0.135, $P$ = 0.026). PLT was associated with male sex ($\beta$ = -0.166, $P$ = 0.006), SGA ($\beta$ = -0.186, $P$ = 0.002) and PIH ($\beta$ = -0.137, $P$ = 0.024).

As shown in Table 3, univariate analysis revealed that MPV was correlated with PROM in late preterm neonates (PROM = 10.1 fL vs. no PROM = 9.8 fL, $P$ = 0.044). Multivariate analysis associated PLT with PIH ($\beta$ = -0.194, $P$ = 0.023), whereas no associations with PCT were found in late preterm neonates.

When all patients were studied, multivariate analysis showed that SGA, male sex and PIH were associated with PCT and PLT (S1 Table). Furthermore, MPV was significantly associated with CAM (CAM = 10.3 fL vs. no CAM = 9.7 fL, $P$ = 0.001) and male sex (male = 9.7 fL vs. female = 9.9 fL, $P$ = 0.011).

Next, we compared the differences of platelet parameters, sex and PIH between SGA and not SGA (S2 Table). S2 Table showed that SGA was significantly associated with PIH in preterm neonates. Furthermore, PCT and PLT were significantly decreased in SGA neonates

**Table 2. Factors affecting platelet parameters in term neonates.**

| (n) | PCT (%) | | PDW (%) | MPV (fL) | PLT (×10³/μL) | |
|---|---|---|---|---|---|---|
| | univariate analysis | multivariate analysis | univariate analysis | univariate analysis | univariate analysis | multivariate analysis |
| | Median [IQR] | p-value (β) | Median [IQR] | Median [IQR] | Median [IQR] | p-value (β) |
| Male (131) | 0.24 [0.20–0.29] | | 10.8 [10.1–12.0] | 9.7 [9.3–10.2] | 25.9 [20.6–29.3] | |
| Female (127) | 0.27 [0.22–0.32] | | 11.1 [10.3–12.5] | 9.8 [9.4–10.4] | 28.8 [22.6–32.8] | |
| | **P = 0.005** | **P = 0.002 (-0.185)** | P = 0.053 | P = 0.139 | **P = 0.009** | **P = 0.006 (-0.166)** |
| RDS (18) | 0.26 [0.22–0.29] | | 10.9 [10.0–13.2] | 9.4 [8.9–10.4] | 26.3 [22.9–34.9] | |
| non-RDS (240) | 0.24 [0.21–0.30] | | 10.9 [10.0–12.3] | 9.8 [9.3–10.3] | 26.6 [21.4–31.3] | |
| | P = 0.720 | | P = 0.842 | P = 0.193 | P = 0.580 | |
| SGA (52) | 0.23 [0.16–0.29] | | 11.0 [10.4–12.5] | 10.0 [9.4–10.4] | 24.8 [17.4–30.5] | |
| non-SGA (206) | 0.26 [0.22–0.30] | | 10.9 [10.9–12.1] | 9.7 [9.3–10.3] | 26.7 [22.3–31.8] | |
| | **P = 0.024** | **P = 0.006 (-0.168)** | P = 0.299 | P = 0.193 | **P = 0.023** | **P = 0.002 (-0.186)** |
| PROM (8) | 0.22 [0.21–0.25] | | 10.3 [9.8–10.9] | 9.4 [9.3–9.8] | 23.8 [20.5–28.1] | |
| non-PROM (250) | 0.26 [0.21–0.30] | | 10.9 [10.2–12.3] | 9.7 [9.3–10.3] | 26.5 [21.5–31.5] | |
| | P = 0.186 | | P = 0.194 | P = 0.286 | P = 0.230 | |
| CAM (9) | 0.21 [0.19–0.29] | | 11.3 [11.0–12.1] | 10.4 [10.1–10.9] | 22.5 [20.1–28.2] | |
| non-CAM (249) | 0.26 [0.21–0.30] | | 10.9 [10.2–12.3] | 9.7 [9.3–10.2] | 26.5 [21.7–31.5] | |
| | P = 0.553 | | P = 0.144 | **P = 0.005** | P = 0.199 | |
| PA (5) | 0.21 [0.19–0.24] | | 11.1 [9.8–14.9] | 10.2 [8.1–10.3] | 23.1 [20.5–27.8] | |
| non-PA (243) | 0.26 [0.21–0.30] | | 10.9 [10.2–12.3] | 9.7 [9.3–10.3] | 26.5 [21.7–31.4] | |
| | P = 0.086 | | P = 0.725 | P = 0.913 | P = 0.319 | |
| PIH (4) | 0.19 [0.12–0.30] | | 11.1 [10.9–12.2] | 9.9 [9.3–10.3] | 17.4 [12.3–19.8] | |
| non-PIH (244) | 0.50 [0.12–0.19] | | 10.9 [10.2–12.3] | 9.7 [9.5–10.2] | 26.5 [21.9–31.4] | |
| | **P = 0.009** | **P = 0.026 (-0.135)** | P = 0.447 | P = 0.683 | **P = 0.007** | **P = 0.024 (-0.137)** |
| GA | r = -0.121 | | r = -0.003 | r = 0.048 | r = -0.072 | |
| | P = 0.052 | | P = 0.958 | P = 0.446 | P = 0.149 | |
| BW | r = 0.070 | | r = -0.053 | r = 0.027 | r = 0.020 | |
| | P = 0.260 | | P = 0.400 | P = 0.666 | P = 0.753 | |
| AP 1 | r = 0.026 | | r = -0.039 | r = -0.052 | r = 0.022 | |
| | P = 0.677 | | P = 0.530 | P = 0.409 | P = 0.730 | |
| AP5 | r = 0.072 | | r = -0.074 | r = -0.048 | r = 0.087 | |
| | P = 0.251 | | P = 0.235 | P = 0.445 | P = 0.165 | |

PCT, platelet count; PDW, platelet distribution width; MPV, mean platelet volume; PLT, platelet count; GA, gestational age; BW, birth weight; RDS, respiratory distress syndrome; SGA, small for gestational age; AP1, Apgar score at 1 min.; AP5, Apgar score at 5 min.; PROM, premature rupture of membranes; CAM, chorioamnionitis; PA, placental abruption; PIH, pregnancy-induced hypertension; IQR, median interquartile range. Significant correlation between GA, BW, Apgar score and platelet parameters were analyzed using Spearman's rank correlation (r). β means standardized regression coefficient.

compared with not SGA neonates. All data pertaining to our analyses are contained within the S1 Text.

## Discussion

This large cohort showed no differences in platelet parameters at birth between late preterm and term neonates. Previous reports indicated that PLT and PCT were lower in late preterm neonates compared with term neonates [12, 17]. Furthermore, they suggested that GA was significantly correlated with PLT, however, in our study, GA did not correlate with PLT [12]. Instead, there was positive correlation between BW and PLT in late preterm neonates by univariate analysis. These differences may be due to differences in sample size and time of

**Table 3. Factors affecting platelet parameters in late preterm neonates.**

| (n) | PCT (%) | | PDW (%) | MPV (fL) | PLT (×10³/µL) | |
|---|---|---|---|---|---|---|
| | univariate analysis Median [IQR] | multivariate analysis p-value (β) | univariate analysis Median [IQR] | univariate analysis Median [IQR] | univariate analysis Median [IQR] | multivariate analysis p-value (β) |
| Male (76) | 0.25 [0.19–0.30] | | 10.9 [9.9–13.5] | 9.8 [9.0–10.2] | 26.9 [22.0–31.7] | |
| Female (66) | 0.25 [0.21–0.31] | | 10.9 [0.3–12.6] | 9.9 [9.5–10.4] | 25.8 [22.6–32.7] | |
| | P = 0.568 | | P = 0.373 | P = 0.167 | P = 0.995 | |
| RDS (13) | 0.22 [0.16–0.26] | | 11.2 [10.0–13.3] | 9.6 [9.0–10.1] | 23.9 [18.6–28.9] | |
| non-RDS (129) | 0.25 [0.21–0.31] | | 10.9 [10.0–12.5] | 9.9 [9.2–10.3] | 27.0 [22.1–32.6] | |
| | P = 0.050 | | P = 0.883 | P = 0.374 | P = 0.138 | |
| SGA (57) | 0.24 [0.17–0.28] | | 10.9 [10.1–13.5] | 9.9 [9.2–10.4] | 24.4 [16.9–29.8] | |
| non-SGA (85) | 0.26 [0.22–0.31] | | 11.0 [10.0–12.5] | 9.8 [9.1–10.2] | 28.0 [23.8–32.8] | |
| | P = 0.020 | P = 0.357 (β = -0.116) | P = 0.533 | P = 0.454 | **P = 0.010** | P = 0.165 (β = -0.172) |
| PROM (33) | 0.25 [0.20–0.32] | | 11.3 [10.5–12.0] | 10.1 [9.8–10.4] | 25.9 [20.1–32.5] | |
| non-PROM (109) | 0.25 [0.20–0.30] | | 10.9 [10.0–13.1] | 9.8 [9.2–10.4] | 26.7 [22.9–32.2] | |
| | P = 0.847 | | P = 0.496 | **P = 0.044** | P = 0.673 | |
| CAM (7) | 0.26 [0.21–0.31] | | 11.2 [10.0–12.0] | 10.3 [9.8–10.6] | 28.0 [24.1–32.4] | |
| non-CAM (135) | 0.25 [0.20–0.31] | | 10.9 [10.0–13.1] | 9.9 [9.0–10.2] | 26.6 [22.1–32.4] | |
| | P = 0.752 | | P = 0.936 | P = 0.087 | P = 0.588 | |
| PA (9) | 0.25 [0.12–0.36] | | 10.1 [10.0–13.5] | 9.8 [9.5–10.6] | 26.6 [11.3–37.2] | |
| non-PA (133) | 0.25 [0.21–0.30] | | 10.9 [10.2–12.5] | 9.9 [9.1–10.3] | 26.7 [22.7–32.0] | |
| | P = 0.937 | | P = 0.566 | P = 0.586 | P = 0.834 | |
| PIH (24) | 0.23 [0.21–0.32] | | 10.9 [9.9–12.2] | 9.9 [9.1–10.3] | 23.7 [15.6–29.1] | |
| non-PIH (118) | 0.25 [0.16–0.27] | | 10.9 [10.0–13.1] | 9.8 [9.4–10.4] | 27.1 [23.1–32.9] | |
| | **P = 0.036** | P = 0.051 (β = -0.170) | P = 0.685 | P = 0.564 | **P = 0.007** | **P = 0.023 (β = -0.194)** |
| GA | r = 0.011 | | r = 0.030 | r = 0.053 | r = 0.026 | |
| | P = 0.895 | | P = 0.722 | P = 0.533 | P = 0.757 | |
| BW | r = 0.180 | | r = -0.040 | r = -0.046 | r = 0.173 | |
| | **P = 0.032** | P = 0.929 (β = -0.011) | P = 0.637 | P = 0.590 | **P = 0.039** | P = 0.960 (β = -0.006) |
| AP 1 | r = 0.028 | | r = 0.115 | r = 0.046 | r = 0.049 | |
| | P = 0.744 | | P = 0.172 | P = 0.583 | P = 0.565 | |
| AP5 | r = -0.002 | | r = 0.117 | r = 0.122 | r = -0.006 | |
| | P = 0.978 | | P = 0.164 | P = 0.149 | P = 0.941 | |

PCT, platelet count; PDW, platelet distribution width; MPV, mean platelet volume; PLT, platelet count; GA, gestational age; BW, birth weight; RDS, respiratory distress syndrome; SGA, small for gestational age; AP1, Apgar score at 1 min.; AP5, Apgar score at 5 min.; PROM, premature rupture of membranes; CAM, chorioamnionitis; PA, placental abruption; PIH, pregnancy-induced hypertension; IQR, median interquartile range. Significant correlation between GA, BW, Apgar score and platelet parameters were analyzed using Spearman's rank correlation (r). β means standardized regression coefficient.

sampling. Other research included 129 neonates: 58 late preterm and 71 full term. Moreover, they found that PLT measured using cord blood was associated with GA and BW in late preterm and term neonates. In our study, we used venous blood at admission to NICU. The mean time of sampling in this study was just under 2 hours after birth.

This study also demonstrates that various maternal and neonatal factors affect platelet parameters in late preterm and term neonates. Especially, late preterm and term SGA neonates

had lower PLT and PCT compared with not SGA neonates as previously described [18, 19]. Lower PLT in SGA indicates the immaturity of thrombopoiesis [20, 21]. Roberts suggested that reduced platelet production was characteristic of preterm neonates and they had fewer circulating megakaryocytes [22]. Another study reported that plasma thrombopoietin concentrations were low in thrombocytopenic newborns, indicative of inadequate up-regulation of TPO production [23]. Platelets are produced in the fetal liver, with production transferred to bone marrow in the third trimester [24]. The liver is one of the first organs to be affected by growth restriction. Furthermore, in this study, PIH was associated with lower PLT and PCT. Moreover, PIH is also a significant risk factor for SGA in this study as previously reported. Maternal PIH can result in neonatal thrombocytopenia [13, 25]. However, the pathogenesis of thrombocytopenia among infants born to mothers with PIH remain unknown. One potential mechanism is that maternal hypertension and resultant fetal hypoxia affects megakaryocyte proliferation [13, 26].

The present study suggested that CAM was associated with MPV in term neonates. However, PROM is not associated with MPV as in a previous study [27]. Recently, a retrospective study has reported significant correlation between MPV and intrauterine infection in neonates with thrombocytopenia and leukopenia at birth [23, 28]. Since CAM was indicative of intrauterine infection, our results are consonant with previous studies. Furthermore, PDW in SGA neonates did not differ from those of not SGA neonates. This result agrees with a previous study [29]. Recently, elevated PDW was associated with poor prognosis of various conditions such as carcinoma and cardiovascular disease [1, 30, 31]. Although, Patrick et al reported that high MPV and PDW showed high specificity in neonates with late sepsis [32], there are no late sepsis cases in this study.

Another important finding of the present study concerns PLT differences by sex. Although there are few reports about the relationships between PLT and sex in neonates, a previous study reported that school-age females had higher PLT than their male counterparts [33]. However, the mechanisms responsible for PLT differences by sex are unknown.

Our study has several limitations. First, we used samples of venous blood on admission. The timing of samples might affect platelet parameters. Second, we could not investigate the relationship between platelet parameters and sepsis because few of our preterm infants had sepsis at birth and we did not investigate changes of platelet parameters during the postnatal period. Some studies associated sepsis with elevated MPV in neonates during the postnatal period [32, 34]. Third, Apgar score was not associated with lower PLT in late preterm and term neonates in this study. Some studies previously showed that low Apgar score at 5 minutes was related to thrombocytopenia [35, 36]. Mario et al showed that SGA, low Apgar score at 5 minutes, cesarean delivery and lower gestational age were risk factors for thrombocytopenia in preterm deliveries between 27 and 35 weeks of gestation [35]. In murine models, it is known that hypoxia affects megakaryocyte progenitors and induces thrombocytopenia [37, 38]. Castle et al reported that severe hypoxia significantly shortened the survival of platelets in a murine model [39]. In this study, there were too few subjects with severe birth asphyxia to make any inferences. Since very low birth weight neonatal platelets were markedly less reactive than adult platelets to ADP/epinephrine, further study to investigate the relationship between severe birth asphyxia requiring epinephrine and PLT reactivity is needed [40]. Finally, the subjects of term neonates admitted to our NICU in this study have various diseases such as transient tachypnea of newborn, pneumothorax and RDS or hypoglycemia or birth asphyxia. Although RDS is not associated with platelet indices in this study, these term neonates strictly differ from healthy term neonates.

In summary, this study demonstrates that various maternal and neonatal factors affect platelet parameters in late preterm and term neonates. In particular, SGA male sex, and PIH

were associated with lower PLT and PCT in late preterm and term neonates. Furthermore, CAM was significantly associated with MPV. Thus, this study demonstrates that different maternal and neonatal complications affect platelet parameters in late preterm and term neonates.

## Supporting information

**S1 Table. Factors affecting platelet parameters in late preterm and term neonates.** (DOCX)

**S2 Table. Platelet parameters in SGA and not neonates.** (DOCX)

**S1 Text. This study's data set.** (XLSX)

## Acknowledgments

We would like to thank Kyohei Miyazaki, Kenichi Sato, Hajime Maeda, and Maki Sato for their productive discussions and comments on the manuscript, as well as for their technical help.

## Author Contributions

**Data curation:** Hayato Go, Nozomi Kashiwabara, Mina Chishiki, Masato Hoshino, Kei Ogasawara.

**Investigation:** Hayato Go.

**Methodology:** Hayato Go.

**Writing – original draft:** Hayato Go.

**Writing – review & editing:** Hitoshi Ohto, Kenneth E. Nollet, Yukihiko Kawasaki, Nobuo Momoi, Mitsuaki Hosoya.

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
