## [Decision Letter · Decision Letter 0]

13 Apr 2020

PONE-D-19-33766

Perinatal factors affecting platelet parameters in late preterm and term neonates.

PLOS ONE

Dear Dr. Hayato Go,

Thank you for submitting your manuscript to PLOS ONE. After careful consideration, we feel that it has merit but does not fully meet PLOS ONE’s publication criteria as it currently stands. Therefore, we invite you to submit a revised version of the manuscript that addresses the points raised during the review process.

Please address the comments identified by the two reviewers at your earliest convenience

We would appreciate receiving your revised manuscript by May 28 2020 11:59PM. To enhance the reproducibility of your results, we recommend that if applicable you deposit your laboratory protocols in protocols.io, where a protocol can be assigned its own identifier (DOI) such that it can be cited independently in the future. For instructions see: http://journals.plos.org/plosone/s/submission-guidelines#loc-laboratory-protocols

We look forward to receiving your revised manuscript.

Kind regards,

Anna Palatnik, M.D.

Academic Editor

PLOS ONE

Journal Requirements:

2. In your ethics statement in the manuscript and in the online submission form, please provide additional information about the patient records used in your retrospective study. Specifically, please ensure that you have discussed whether all data were fully anonymized before you accessed them and/or whether the IRB or ethics committee waived the requirement for informed consent. If patients provided informed written consent to have data from their medical records used in research, please include this information.

Please also clarify whether blood collection from the umbillical cord blood or peripheral venipuncture was a part of routine and standard care or whether it was collected specifically for this study.

3. Thank you for stating the following financial disclosure:"NO - Include this sentence at the end of your statement: The funders had no role in study design, data collection and analysis, decision to publish, or preparation of the manuscript."

Please provide an amended Funding Statement that declares *and fully names all* the funding or sources of support received during this specific study (whether external or internal to your organization) as detailed online in our guide for authors at http://journals.plos.org/plosone/s/submit-now.  

Additional Editor Comments (if provided):

Reviewer 1:

Authors have chosen interesting subject for research.

However, there are many unanswered questions in the mansucript-

1. The study is retrospective in nature; what were the indications for doing all the studied platelet parameters in these late preterm and term neonates at the first place? Is that a routine practice in their unit to screen all babies for platelet issues? Is it justified ethically?

2.All late preterms and term neonates admitted during the study period (491 total) were included in the study and there is no exclusion.How is the data managed in the unit?

3. The data shown in table 1 describes platelet parameters between late PT and term neonates. The given data is highly inadequate. There is no mention whether it is mean or median and no standard deviation or IQR is shown.Further except for GA, BW and PCT , other values are exactly the same which further raise queries on data collection and analysis.

4.Again in table 2 and 3, there is no mention of measures of variation.In multivariate analysis , odds ratio should have been shown instead of just p value.

5.Authors mention use of cord blood or venous blood at admission for study.What site of cord blood was used whether maternal side or neonatal side?Further the mean sampling time was under 2 hours , the studied parameters might reflect maternal characteristics. It would have been clinically relevant if postnatal changes also studied simultaneously.

6.Authors rightly acknowledged their limitations about sepsis and its effect on platelet parameters, which is a major factor which affect platelets

7.Typographical error in the introduction first line -"....in various disease diseases"; remove disease.

Reviewer 2:

General Comments:

This manuscript attempts to provide a cross sectional observation of the platelet’s morphology parameters associated with late preterm and term births. While compelling, the manuscript is lacking in both writing and grammatical issues as well as is lacking in some important analysis. Overall if these issues are addressed this could be an important contribution to the base of knowledge in neonatal intensive care.

The most obvious omission is the lack of connection between the existing base of knowledge on this topic om platelet function and how this would attribute to a clinical outcome. What is the hypothesis? How would a change in platelet number or size influence clinical care?

Better definition of clinical variables needs to be presented. The AGA and SGA descriptors need to be updated. It is important to note in the demographic table 1, what is the proportion of SGA’s in the term and late preterm population.

Specific comments follow below:

Page: 12

1. The opening sentence of the Introduction is very vague and not compelling. Please be more specific and deliberate in your writing.

2. What is your hypothesis? It’s not clear what you are adding to the existing body of knowledge.

3. What factors? What comparisons? What are the variables you are interested in studying? At this point, this sounds like an observational study, so please state that explicitly.

4. Why was analysis on preterm babies not performed? That data is discussed in the introduction, but you do not provide a rationale for why it is not included in this study. Why did you limit to term and late preterm?

Page 13:

1. When were these samples collected? Beginning of Day of Life 1? At the end? Any time in the first 24 hours? This will confound your results significantly.

2. One of the factors you have failed to examine is how many babies required resuscitation at birth with epinephrine? Epinephrine has been reported to cause platelet aggregation as well as recruitment into the circulation. See Kjeldsen SE, Weder AB, Egan B, Neubig R, Zweifler AJ, Julius S. Effect of circulating epinephrine on platelet function and hematocrit. Hypertension. 1995;25(5):1096–1105.

3. Furthermore, reactivity to epinephrine via ADP and thromboxane differs between neonates and adults. Rajasekhar D, Barnard MR, Bednarek FJ, Michelson AD. Platelet hyporeactivity in very low birth weight neonates. Thromb Haemost. 1997;77(5):1002–1007.

4. Therefore, it is important to note what the exposure to resuscitation and epinephrine is in these subjects.

Page 14:

1. It would have been nice if platelet morphology was examined via microscopy on peripheral smears

Page 15:

1. Here it would be very important to separate the term and late preterm subjects by AGA and SGA status. So, have four columns instead of two.

Page 17:

1. What is your definition of SGA?

Page 18:

1. You have omitted definitions for several of these variables. Is CAM clinical or culture proven? What is the definition of clinical chorio at your institution?

Page 24:

1. Timing of sampling needs to be clearly stated in the methods.

2. You also need to differentiate which values were collected from cord and which were from venous samples. Were these values compared to one another? If not, it would be vital to know if there is a difference in parameters from these two methods of sampling.

Page 25:

1. Determining exposure to epinephrine and resuscitation may help understand this better. See comment above.

Reviewers' comments:

Reviewer's Responses to Questions

**Comments to the Author**

1. Is the manuscript technically sound, and do the data support the conclusions?

Reviewer #1: Partly

Reviewer #2: No

2. Has the statistical analysis been performed appropriately and rigorously? 

Reviewer #1: Yes

Reviewer #2: No

3. Have the authors made all data underlying the findings in their manuscript fully available?

Reviewer #1: Yes

Reviewer #2: No

4. Is the manuscript presented in an intelligible fashion and written in standard English?

Reviewer #1: No

Reviewer #2: Yes

5. Review Comments to the Author

Reviewer #1: General Comments:

This manuscript attempts to provide a cross sectional observation of the platelet’s morphology parameters associated with late preterm and term births. While compelling, the manuscript is lacking in both writing and grammatical issues as well as is lacking in some important analysis. Overall if these issues are addressed this could be an important contribution to the base of knowledge in neonatal intensive care.

The most obvious omission is the lack of connection between the existing base of knowledge on this topic om platelet function and how this would attribute to a clinical outcome. What is the hypothesis? How would a change in platelet number or size influence clinical care?

Better definition of clinical variables needs to be presented. The AGA and SGA descriptors need to be updated. It is important to note in the demographic table 1, what is the proportion of SGA’s in the term and late preterm population.

Specific comments follow below:

Page: 12

1. The opening sentence of the Introduction is very vague and not compelling. Please be more specific and deliberate in your writing.

2. What is your hypothesis? It’s not clear what you are adding to the existing body of knowledge.

3. What factors? What comparisons? What are the variables you are interested in studying? At this point, this sounds like an observational study, so please state that explicitly.

4. Why was analysis on preterm babies not performed? That data is discussed in the introduction, but you do not provide a rationale for why it is not included in this study. Why did you limit to term and late preterm?

Page 13:

1. When were these samples collected? Beginning of Day of Life 1? At the end? Any time in the first 24 hours? This will confound your results significantly.

2. One of the factors you have failed to examine is how many babies required resuscitation at birth with epinephrine? Epinephrine has been reported to cause platelet aggregation as well as recruitment into the circulation. See Kjeldsen SE, Weder AB, Egan B, Neubig R, Zweifler AJ, Julius S. Effect of circulating epinephrine on platelet function and hematocrit. Hypertension. 1995;25(5):1096–1105.

3. Furthermore, reactivity to epinephrine via ADP and thromboxane differs between neonates and adults. Rajasekhar D, Barnard MR, Bednarek FJ, Michelson AD. Platelet hyporeactivity in very low birth weight neonates. Thromb Haemost. 1997;77(5):1002–1007.

4. Therefore, it is important to note what the exposure to resuscitation and epinephrine is in these subjects.

Page 14:

1. It would have been nice if platelet morphology was examined via microscopy on peripheral smears

Page 15:

1. Here it would be very important to separate the term and late preterm subjects by AGA and SGA status. So, have four columns instead of two.

Page 17:

1. What is your definition of SGA?

Page 18:

1. You have omitted definitions for several of these variables. Is CAM clinical or culture proven? What is the definition of clinical chorio at your institution?

Page 24:

1. Timing of sampling needs to be clearly stated in the methods.

2. You also need to differentiate which values were collected from cord and which were from venous samples. Were these values compared to one another? If not, it would be vital to know if there is a difference in parameters from these two methods of sampling.

Page 25:

1. Determining exposure to epinephrine and resuscitation may help understand this better. See comment above.

Reviewer #2: Authors have chosen interesting subject for research.

However, there are many unanswered questions in the mansucript-

1. The study is retrospective in nature; what were the indications for doing all the studied platelet parameters in these late preterm and term neonates at the first place? Is that a routine practice in their unit to screen all babies for platelet issues? Is it justified ethically?

2.All late preterms and term neonates admitted during the study period (491 total) were included in the study and there is no exclusion.How is the data managed in the unit?

3. The data shown in table 1 describes platelet parameters between late PT and term neonates. The given data is highly inadequate. There is no mention whether it is mean or median and no standard deviation or IQR is shown.Further except for GA, BW and PCT , other values are exactly the same which further raise queries on data collection and analysis.

4.Again in table 2 and 3, there is no mention of measures of variation.In multivariate analysis , odds ratio should have been shown instead of just p value.

5.Authors mention use of cord blood or venous blood at admission for study.What site of cord blood was used whether maternal side or neonatal side?Further the mean sampling time was under 2 hours , the studied parameters might reflect maternal characteristics. It would have been clinically relevant if postnatal changes also studied simultaneously.

6.Authors rightly acknowledged their limitations about sepsis and its effect on platelet parameters, which is a major factor which affect platelets

7.Typographical error in the introduction first line -"....in various disease diseases"; remove disease.

6. PLOS authors have the option to publish the peer review history of their article (what does this mean?). If published, this will include your full peer review and any attached files.

Reviewer #1: No

Reviewer #2: No

---

## [Author Response · Author response to Decision Letter 0]

28 Jul 2020

June 9, 2020

Anna Palatnik, MD

RE:　Revised Manuscript PONE-D-19-33766

Title: "Perinatal factors affecting platelet parameters in late preterm and term neonates."

Dear Dr. Palatnik,

Thank you for your letter of April 13, 2020 regarding manuscript PONE-D-19-33766. We appreciate the opportunity to respond to reviewers’ comments, and thank everyone for insightful guidance that has helped us significantly improve the paper.

Below, please find our responses to peer review, incorporated by reference into this cover letter, including required text pertaining to financial disclosure.

It will be a privilege to earn your further consideration toward the publication of this manuscript.

Sincerely,

Hayato Go, MD, PhD

Department of Pediatrics, Fukushima Medical University School of Medicine, Fukushima, 960-1295, Japan

 

#Editor’s comments

Response

Thank you for your suggestion. We revised the manuscript to meet PLOS ONE’s style requirements.

2. In your ethics statement in the manuscript and in the online submission form, please provide additional information about the patient records used in your retrospective study. Specifically, please ensure that you have discussed whether all data were fully anonymized before you accessed them and/or whether the IRB or ethics committee waived the requirement for informed consent. If patients provided informed written consent to have data from their medical records used in research, please include this information.

Please also clarify whether blood collection from the umbilical cord blood or peripheral venipuncture was a part of routine and standard care or whether it was collected specifically for this study.

Response

Thank you for your suggestion.

Following this guidance, we added information about patient records in Supplemental Table 1.

In our NICU, complete blood count (CBC) is routinely performed on admission. However, most of the samples were from peripheral venipuncture and only two samples were from umbilical cord blood. We removed the cases using umbilical cord blood.

3. Thank you for stating the following financial disclosure: "NO - Include this sentence at the end of your statement: The funders had no role in study design, data collection and analysis, decision to publish, or preparation of the manuscript."

a. Please provide an amended Funding Statement that declares *and fully names all* the funding or sources of support received during this specific study (whether external or internal to your organization) as detailed online in our guide for authors at http://journals.plos.org/plosone/s/submit-now. 

Response

Thank you, we attest to the following as true:

“This research proceeded without the benefit of grant money or any other external financial support. Accordingly, funders had no role in study design, data collection and analysis, decision to publish, or preparation of the manuscript.”

Response

Thank you for your suggestion.

Following this guidance, we added information about patient records in Supporting Table 1.

Response

Thank you for your suggestion.

Following this guidance, we added information about patient records in Supporting Table 1. We added the mean time of sampling after birth in the results (new Fig 2) and deleted the words “data not shown” from our discussion.

We also added the following sentence in results section.

“The median sampling time was 82 min as shown in Fig 2.”

Reviewer 1:

Authors have chosen interesting subject for research.

However, there are many unanswered questions in the mansucript-

1. The study is retrospective in nature; what were the indications for doing all the studied platelet parameters in these late preterm and term neonates at the first place? Is that a routine practice in their unit to screen all babies for platelet issues? Is it justified ethically?

Response

Thank you for these questions.

In our NICU, complete blood count (CBC) is routinely performed on all babies upon admission. Most late preterm and term neonates admitted to our NICU have various disease such as transient tachypnea of the newborn, pneumothorax, and respiratory distress syndrome, hypoglycemia, or birth asphyxia, etc. So, these term neonates differ from healthy term neonates. Ethically, we are trying to exercise an abundance of caution by collecting data that might guide and improve patient care.

Therefore, we also added the following sentence in discussion.

“Finally, term neonates admitted to our NICU in this study have various diseases such as transient tachypnea of the newborn, pneumothorax, RDS, hypoglycemia, or birth asphyxia. Although RDS is not associated with platelet indices in this study, these term neonates differ from healthy term neonates.”

2.All late preterm and term neonates admitted during the study period (491 total) were included in the study and there is no exclusion. How is the data managed in the unit?

Response

Thank you. We added a flow chart indicative of our data management. We found that the original data set included some cases such as congenital heart disease and chromosomal abnormalities. After excluding these, the total number of subjects decreased from 491 to 400. We reanalyzed the data using these 400 total neonates.

The flow chart appears in new Fig 1.

3. The data shown in table 1 describes platelet parameters between late PT and term neonates. The given data is highly inadequate. There is no mention whether it is mean or median and no standard deviation or IQR is shown. Further except for GA, BW and PCT, other values are exactly the same which further raise queries on data collection and analysis.

Response

We agree with reviewer’s comment.

We reanalyzed the data in Table 1 and added IQR. 

4.Again in table 2 and 3, there is no mention of measures of variation. In multivariate analysis, odds ratio should have been shown instead of just p value.

Response

We agree with reviewer’s comment.

We reanalyzed the data in Table 2 and 3 and added IQR in each Table. Furthermore, in terms of multivariate analysis, we used multiple regression analysis. Since platelet parameters were continuous variables, we mentioned both of p value and standardized coefficients. We are open to further advice about the necessity of 95% Confidence interval for unstandardized coefficients. 

5.Authors mention use of cord blood or venous blood at admission for study. What site of cord blood was used whether maternal side or neonatal side? Further the mean sampling time was under 2 hours , the studied parameters might reflect maternal characteristics. It would have been clinically relevant if postnatal changes also studied simultaneously.

Response

We agree with reviewer’s comment.

When collecting cord blood, we typically use the neonatal side. However, in this study, there were only two samples using cord blood. So, we deleted these samples from our analysis.

On the other hand, we reanalyzed the sampling time. The median of sampling time was 82.0 min (IQR 50-120). Data was shown in new Fig 2. The data discussed in this study have been deposited in Supplemental Table 1.

6.Authors rightly acknowledged their limitations about sepsis and its effect on platelet parameters, which is a major factor which affect platelets

Response

We agree with reviewer’s comment.

In this study, we could not find out which factors of SGA, PIH, and sepsis affect platelet parameters. While we are at a loss to explain this, we suspect that SGA, PIH, and sepsis are associated with platelet parameters via various mechanisms.

7.Typographical error in the introduction first line -"....in various disease diseases"; remove disease.

Response

Thank you, we removed the sentence.

Reviewer 2:

General Comments: 

This manuscript attempts to provide a cross sectional observation of the platelet’s morphology parameters associated with late preterm and term births. While compelling, the manuscript is lacking in both writing and grammatical issues as well as is lacking in some important analysis. Overall if these issues are addressed this could be an important contribution to the base of knowledge in neonatal intensive care. 

The most obvious omission is the lack of connection between the existing base of knowledge on this topic om platelet function and how this would attribute to a clinical outcome. What is the hypothesis? How would a change in platelet number or size influence clinical care?

Response

Thank you, yes, we are eager to advance knowledge to improve neonatal care. This includes sharing data for which explanations might ultimately emerge from other groups looking at our data in the context of their own research. We have added text and references to put this research in a broader context of existing knowledge. In addition to this and other revisions suggested by the editor and reviewers, our co-author whose first language is English has indeed noticed and corrected some linguistic irregularities that were in the original submission. 

Specific comments follow below:

Page: 12

1. The opening sentence of the Introduction is very vague and not compelling. Please be more specific and deliberate in your writing.

Response

Thank you for the excellent suggestion. We added text to the beginning and end of the first paragraph of our introduction.

2. What is your hypothesis? It’s not clear what you are adding to the existing body of knowledge.

Response

Thank you. Now, in the last sentence of the second paragraph or our introduction, we hypothesize that “that platelet parameters such as MPV, PCT, and PDW could be affected by perinatal factors. The objective of this study was to investigate the factors affecting platelet parameters at birth in late preterm and term neonates.”

3. What factors? What comparisons? What are the variables you are interested in studying? At this point, this sounds like an observational study, so please state that explicitly.

Response

Thank you, this, too has been incorporated into our revised first paragraph.

4. Why was analysis on preterm babies not performed? That data is discussed in the introduction, but you do not provide a rationale for why it is not included in this study. Why did you limit to term and late preterm?

Response

Thank you for these questions. We previously reported the association between very low birth weight and platelet parameters using the data collected from 2006 to 2017. So, we added the sentence of our previous study in the introduction, below.

“We previously reported that higher MPV correlates with mortality among those born at <32 weeks’ gestation (11).”

Page 13: 

1. When were these samples collected? Beginning of Day of Life 1? At the end? Any time in the first 24 hours? This will confound your results significantly.

Response

Thank you. We collected these samples at admission, with a time distribution described in the text and in the new Figure 2.

2. One of the factors you have failed to examine is how many babies required resuscitation at birth with epinephrine? Epinephrine has been reported to cause platelet aggregation as well as recruitment into the circulation. See Kjeldsen SE, Weder AB, Egan B, Neubig R, Zweifler AJ, Julius S. Effect of circulating epinephrine on platelet function and hematocrit. Hypertension. 1995;25(5):1096–1105.

Response

Thank you for this reference. Among our cases, two babies required resuscitation at birth with epinephrine. Although we could not investigate the correlation between epinephrine and platelet count, in these two cases we measured 22.9 ×103/μL and 24.5 ×103/μL.

Furthermore, we found that some cases included congenital anomalies such as congenital heart disease and chromosomal abnormalities. After excluding these neonates (new Figure 1), the total number of patients decreased from 491 to 400. After reanalyzing the data using just these 400, Apgar score was not associated with platelet parameters. This is possibly because severe birth asphyxia cases were decreased. 

We added to the discussion as follows:

“Third, Apgar score was not associated with lower PLT in late preterm and term neonates in this study. Some studies previously showed that low Apgar score at 5 minutes was related to thrombocytopenia (35,36). Mario et al showed that SGA, low Apgar score at 5 minutes, cesarean delivery and lower gestational age were risk factors for thrombocytopenia in preterm deliveries between 27 and 35 weeks of gestation (35). In murine models, it is known that hypoxia affects megakaryocyte progenitors and induces thrombocytopenia (37,38). Castle et al reported that severe hypoxia significantly shortened the survival of platelets in a murine model (39). In this study, there were too few subjects with severe birth asphyxia to make any inferences. Since very low birth weight neonatal platelets were markedly less reactive than adult platelets to ADP/epinephrine, further study to investigate the relationship between severe birth asphyxia requiring epinephrine and PLT reactivity is needed (40).” 

3. Furthermore, reactivity to epinephrine via ADP and thromboxane differs between neonates and adults. Rajasekhar D, Barnard MR, Bednarek FJ, Michelson AD. Platelet hyporeactivity in very low birth weight neonates. Thromb Haemost. 1997;77(5):1002–1007. Therefore, it is important to note what the exposure to resuscitation and epinephrine is in these subjects.

Thank you for the citations and guidance. Pertinent to this is the new final sentence in the text above, repeated here: “Since very low birth weight neonatal platelets were markedly less reactive than adult platelets to ADP/epinephrine, further study to investigate the relationship between severe birth asphyxia requiring epinephrine and PLT reactivity is needed (40).”

Page 14:

1. It would have been nice if platelet morphology was examined via microscopy on peripheral smears.

Response

We agree, but did not assess the platelet morphology, and so we cannot add any associations between platelet morphology and other parameters. We hope to do this in the future, and hope also to see such data from other investigators.

Page 15: 

1. Here it would be very important to separate the term and late preterm subjects by AGA and SGA status. So, have four columns instead of two.

Response

We agree, and reanalyzed the data after separating the term and preterm subjects by SGA and AGA (re-branded as “not SGA”), now in revised tables.

Page 17: What is your definition of SGA? Better definition of clinical variables needs to be presented. The AGA and SGA descriptors need to be updated. It is important to note in the demographic table 1, what is the proportion of SGA’s in the term and late preterm population. 

Response

Thank you for your suggestion. We added the definition of SGA in Methods as below.

“Infants whose birth weight and height were below the 10th percentile of the normal curve at each GA were classified as SGA.”

Furthermore, we changed the word “AGA” to “not SGA”.

Page 18:

1. You have omitted definitions for several of these variables. Is CAM clinical or culture proven? What is the definition of clinical chorio at your institution?

Response

Thank you. We followed standard clinical diagnostic criteria for CAM (Tita AT, Andrew WW. Diagnosis and management of clinical chorioamnionitis. Clinics in Perinatology. 2010;37(2):339-354), with histopathological confirmation in the case of preterm births, for which placentas are routinely sent to pathology. Cultures were not performed to diagnose CAM. Text has been added to the “Prenatal and postnatal risk factors” subsection of our Methods.

Page 24:

1. Timing of sampling needs to be clearly stated in the methods.

Response

Thank you for your suggestion. We added the mean sampling time in Fig 2.

2. You also need to differentiate which values were collected from cord and which were from venous samples. Were these values compared to one another? If not, it would be vital to know if there is a difference in parameters from these two methods of sampling.

Response

Thank you for your suggestion. We added the mean sampling time in Methods. In this study, there were only two samples of cord blood. So, we deleted these two samples from this study.

Page 25:

1. Determining exposure to epinephrine and resuscitation may help understand this better. See comment above.

Response

Thank you for your suggestion. We added text, as noted above.

---

## [Decision Letter · Decision Letter 1]

21 Oct 2020

PONE-D-19-33766R1

Perinatal factors affecting platelet parameters in late preterm and term neonates.

PLOS ONE

Dear Dr. Go,

Thank you for submitting your manuscript to PLOS ONE. After careful consideration, we feel that it has merit but does not fully meet PLOS ONE’s publication criteria as it currently stands. Therefore, we invite you to submit a revised version of the manuscript that addresses the points raised during the review process.

Please justify why the sample size was reduced to 400 from the original 491 in the first submission.

*PLOS ONE* does not copyedit accepted manuscripts, so the language in submitted articles must be clear, correct, and unambiguous. We recommend that authors seek independent editorial help before submitting a revision. These services can be found on the web using search terms like “scientific editing service” or “manuscript editing service.”

We look forward to receiving your revised manuscript.

Kind regards,

Kelli K Ryckman

Academic Editor

PLOS ONE

Reviewers' comments:

Reviewer's Responses to Questions

**Comments to the Author**

1. If the authors have adequately addressed your comments raised in a previous round of review and you feel that this manuscript is now acceptable for publication, you may indicate that here to bypass the “Comments to the Author” section, enter your conflict of interest statement in the “Confidential to Editor” section, and submit your "Accept" recommendation.

Reviewer #1: All comments have been addressed

Reviewer #2: (No Response)

2. Is the manuscript technically sound, and do the data support the conclusions?

Reviewer #1: Yes

Reviewer #2: Partly

3. Has the statistical analysis been performed appropriately and rigorously? 

Reviewer #1: Yes

Reviewer #2: (No Response)

4. Have the authors made all data underlying the findings in their manuscript fully available?

Reviewer #1: Yes

Reviewer #2: Yes

5. Is the manuscript presented in an intelligible fashion and written in standard English?

Reviewer #1: Yes

Reviewer #2: No

6. Review Comments to the Author

Reviewer #1: Thank you for addressing the comments/reviews you received. Most issues have been adequately addressed. My only remaining comment is that the results section is still difficult to follow with the abundance of abbreviations being used. Table 2 is also very confusing. Perhaps by reversing the X and Y axis, it will be easier to read and follow. This also applies to Table 3 and 4.

Reviewer #2: 1.Although , the authors explain that sampling was done in order to exercise caution , it is still unnecessary to do samplings routinely in newborn and the practice is to avoid unless indicated in order to avoid sampling induced blood loss and consequent anemia.

2. The flow chart has probably typographical error with regards to defining premature birth as > /= 35 weeks .Needs clarification on below which gestation they have defined as premature birth.

3.Regarding the statistical analysis , the authors need to compile the tables more compressed and crisp so that it is reading friendly. Emphasis should be given to what does it imply . The number of tables should be restricted to 2 or 3 maximum.Rest statistical tables can be added as supplementary material

4.Odds ratio may be preferred with 95% CI for interpretation of regression analysis

5.If a complete blood count has also been done along with platelets , then including TLC and ANC as a proxy for sepsis and correlating it with platelet parameters could be of value.

7. PLOS authors have the option to publish the peer review history of their article (what does this mean?). If published, this will include your full peer review and any attached files.

Reviewer #1: No

Reviewer #2: No

---

## [Author Response · Author response to Decision Letter 1]

3 Nov 2020

Nov 3, 2020

Kelli K Ryckman

Editor, PLoS One

RE:　Revised Manuscript PONE-D-19-33766

Title: " Perinatal factors affecting platelet parameters in late preterm and term neonates.

 "

Dear Kelli K Ryckman

Thank you for your letter of Oct 21, 2020 regarding our manuscript- PONE-D-19-33766. We appreciate the opportunity to respond to the reviewer’s comments.

We wish to express our appreciation to the reviewers and editor for their insightful comments,

which have helped us significantly improve the paper.

We've checked your submission and before we can proceed, we need you to address the following issues:

1) Please justify why the sample size was reduced to 400 from the original 491 in the first submission.

Response

Thank you. As we mentioned in the last revision, we found that the original data set included some cases such as congenital heart disease and chromosomal abnormalities. After excluding these cases, the total number of subjects decreased from 491 to 400. 

Reviewer #1: Thank you for addressing the comments/reviews you received. Most issues have been adequately addressed. 

My only remaining comment is that the results section is still difficult to follow with the abundance of abbreviations being used. Table 2 is also very confusing. Perhaps by reversing the X and Y axis, it will be easier to read and follow. This also applies to Table 3 and 4.

Response

Thank you for your suggestion. 

We revised Table 2,3,4 so that it will be easier to read and follow.

Reviewer #2: 

1.Although, the authors explain that sampling was done in order to exercise caution, it is still unnecessary to do samplings routinely in newborn and the practice is to avoid unless indicated in order to avoid sampling induced blood loss and consequent anemia.

Response

Thank you for this comment.

We concur with modern principles of patient blood management. A complete blood count (CBC) is routinely performed on all admissions to NICU, because all of these late preterm and term neonates have various diseases such as jaundice, transient tachypnea of the newborn, pneumothorax, respiratory distress syndrome, hypoglycemia, birth asphyxia, etc. NICU neonates differ from healthy term neonates. Ethically, our abundance of caution includes using blood samples that need to be drawn for immediate NICU care to also collect data that might guide and improve future patient care. 

Therefore, we also added the following sentence in discussion. 

“Finally, term neonates admitted to our NICU in this study have various diseases such as transient tachypnea of the newborn, pneumothorax, RDS, hypoglycemia, or birth asphyxia. Although RDS is not associated with platelet indices in this study, these term neonates differ from healthy term neonates.” 

2. The flow chart has probably typographical error with regards to defining premature birth as > /= 35 weeks .Needs clarification on below which gestation they have defined as premature birth.

Response

Thank you for your suggestion. We revised it.

3.Regarding the statistical analysis , the authors need to compile the tables more compressed and crisp so that it is reading friendly. Emphasis should be given to what does it imply . The number of tables should be restricted to 2 or 3 maximum.Rest statistical tables can be added as supplementary material

Response

Thank you for your suggestion.

We revised the Tables and convert previous Table 4 and Table 5 to supplemental Table 1 and supplemental Table 2.

4.Odds ratio may be preferred with 95% CI for interpretation of regression analysis

Response

Thank you for your suggestion.

Although we could, in principle, add 95% CI in the Table, we have to maintain readability, so in practice, we do not include it.

5.If a complete blood count has also been done along with platelets, then including TLC and ANC as a proxy for sepsis and correlating it with platelet parameters could be of value.

Response

While agreeing in principle, we encountered few sepsis cases at birth. Therefore, we could not assess any correlations between platelet parameters and sepsis.

Thank you again for reconsidering our manuscript for publication. We greatly appreciate the reviewers’ comments and suggestions. We are confident that the paper is now greatly improved and we hope that it is now acceptable for publication.

Sincerely,

Hayato Go, MD, PhD

Pediatrics School of Medicine

Fukushima Medical University

1st Hikarigaoka

Fukushima Fukushima Japan, 960-1295

phone#: 81-24-547-1295

FAX#: 81-24-548-6578

E-mail: go@fmu.ac.jp

---

## [Editor Report · Decision Letter 2]

5 Nov 2020

Perinatal factors affecting platelet parameters in late preterm and term neonates.

PONE-D-19-33766R2

Dear Dr. Go,

We’re pleased to inform you that your manuscript has been judged scientifically suitable for publication and will be formally accepted for publication once it meets all outstanding technical requirements.

Kind regards,

Kelli K Ryckman

Academic Editor

PLOS ONE
---

## [Editor Report · Acceptance letter]

9 Nov 2020

PONE-D-19-33766R2 

Perinatal factors affecting platelet parameters in late preterm and term neonates. 

Dear Dr. Go:

I'm pleased to inform you that your manuscript has been deemed suitable for publication in PLOS ONE. Congratulations! Your manuscript is now with our production department. 

Kind regards, 

on behalf of

Dr. Kelli K Ryckman 

Academic Editor

PLOS ONE